# Factors Influencing Purchase Intention of Food Surplus through a Food-Sharing Platform

**Nan Hua** [1], **Randall Shannon** [2,*], **Murtaza Haider** [2] and **George P. Moschis** [2]

1 College of Management, Mahidol University, Bangkok 10400, Thailand; beckhamhua8507@gmail.com
2 Center for Research on Sustainable Leadership, College of Management, Mahidol University, Bangkok 10400, Thailand; ghandharires@gmail.com (M.H.); gmoschis@gsu.edu (G.P.M.)
* Correspondence: a.randall@gmail.com

**Abstract:** Food waste is a serious issue around the world. One way to address this issue is distributing food surpluses through food-sharing platforms. There are a limited number of empirical studies investigating the drivers of using food surplus-sharing platforms, particularly in developing countries. This paper investigates the impacts and connections between environmental concern, perceived playfulness, social norms, food waste awareness, price consciousness, food neophobia, and purchase intention of food surplus through a food-sharing platform in Thailand. A sample of 284 Yindii users was analyzed by using exploratory factor analysis and multiple regression. Empirical results revealed environmental concern and perceived playfulness to be the primary constructs influencing consumers' purchase intention toward food surplus. The results suggest that perceived playfulness is the most crucial determinant affecting purchase intention. Our results also indicated people who have obtained a higher education level and the low-income group show a higher purchase intention toward food surplus products. This research is the first attempt to study food surplus redistribution in Thailand. This study contributes to the literature and provides insights for practitioners with several implications.

**Keywords:** food surplus; food-sharing platform; purchase intention; environmental concern; perceived playfulness; sustainable consumption

## 1. Introduction

Food is the most essential product in our daily life. Although food processing and production have increased with developing technologies, a huge number of people still live in hunger and are undernourished internationally. Around 820 million people continue to live in hunger every day [1], and the number of undernourished populations may surpass 840 million in 2030 if the trend does not change [2]. However, roughly one-third of the world's food is lost or wasted every year [1]. Global average food waste is around 121 kg per capita per year [3]. In 2019, approximately 381 million undernourished people resided in Asia, accounting for over 50 percent of the global total undernourished population [2]. Around 9.8 percent of the population is undernourished in South East Asian countries, and 9.3 percent of the population of 6.5 million Thais live in a malnutrition situation [2]. In addition, ASEAN countries experience a high level of post-harvest losses, with estimated losses in rice of up to 27 percent and losses in fruit and vegetables of 20 percent [4].

Food waste is one of the major challenges in our world [1]. Food waste appears at different stages in the food supply chain, and it is defined differently by researchers [5]. In 1981, the FAO [6] defined food waste as wholesome edible material intended for human consumption arising at any point in the food supply chain that is, instead, discarded, lost, degraded, or consumed by pests. Based on the FAO's definition, Stuart [7] adds that food waste should also include edible material that is intentionally fed to animals or is a food-processing byproduct removed from human food. In addition to the above definitions,

from the nutrition aspect, Smil [8] adds the energy value gap between consumed food per capita and needed food per capita. In its updated report in 2019, the FAO [1] defined food waste as the decrease in quantity or quality of food at the retail and consumption level.

Food waste has a significant negative impact on the environment, society, and economy [4,9,10]. The total cost of food losses and waste is around USD 2.6 trillion per year, which roughly equals the GDP of France [11]. The annual economic costs of food loss and waste are significant and reached around USD 1 trillion [11]. In addition to the economic costs of food losses and waste, food loss and waste also contribute to broader social costs that have impacts on people's well-being and health. The social costs introduced by food loss and waste are around USD 900 billion [11]. Furthermore, food loss and waste have a severe impact on the environment. Every year, global food loss and waste create approximately 8 percent of greenhouse gas emissions, which equals around USD 940 billion. After the United States of America and China, global food loss and waste rank as the third largest emitter of greenhouse gas [4]. Papargyropoulou et al. [12] proposed a five-level food waste hierarchy framework and identified that prevention is the most effective approach, followed by food surplus redistribution, to tackle global food loss and waste problems.

## 2. Research Background

Food surplus covers a narrower scope than food waste. Papargyropoulou et al. [12] define agricultural output or the amount of food produced exceeding human needs as food surplus. UNEP [3] refers to food surplus as "food that is redistributed for consumption by people, used for animal feed or used for bio-based materials/biochemical processing". Facchini et al. [13] describe food surplus as food that is completely edible and reusable but is discarded by producers and retailers due to aesthetic reasons or low demand. Teigiserova, Hamelin, and Thomsen [10] offer a narrower scope of the term that only includes the nutritional surplus of food that is fit for human consumption. In this research, food surplus is defined as food that can be redistributed for human consumption from the human food supply chain at food retail and food service levels. In the literature, food surplus and surplus food are used interchangeably, so this research will use food surplus as a term representing both [3,10,12,14,15].

Food surplus redistribution is promoted as an effective approach and preferred option to reduce food waste [16]. Food surplus redistribution has increased rapidly in some developed countries. In 2018, food surplus redistribution had increased by 96 percent since 2015, or an extra USD 103 million worth of food, or an additional 65 million meals annually in the UK [15]. Food surplus redistribution organizations can be categorized into two groups: commercial redistribution organizations, businesses that redistribute food surplus for profit; and charitable and social redistribution organizations, organizations that redistribute food surplus for social and environmental reasons [15]. Recently published research investigating Japanese food sharing [17] and food waste in Bangkok [18] has recommended investigation on online food-sharing platforms. Nevertheless, research on food waste issues is scarce in developing countries [19]. Although a few studies explored consumers' purchase intention of food surplus through food-sharing platforms, our understanding of this topic is still limited [20], specifically in developing countries [21]. Empirical studies on drivers of using surplus food-sharing platforms are scarce [22]. There is no study about food surplus redistribution and food-sharing platforms in Thailand.

With digital technology and the emergence of the sharing economy, several food-sharing or redistribution platforms and applications have been developed to tackle the food waste issue, such as Flashfood in Canada [23], OLIO in Italy [22], Karma in Nordic countries [24], Too Good To Go in the UK [25], Needy Serve in Bangladesh [26], Eat 'N Save in Colombia [27], and so forth. Papargyropoulou, Fearnyough, Spring, and Antal [14] find that a technology platform is one of the effective models for food surplus redistribution. Harvey et al. [28] demonstrate that the food-sharing application is a determining factor for food redistribution to happen. Apostolidis, Brown, Wijetunga, and Kathriarachchi [21] support that food waste mobile applications allow companies to make money while dis-

tributing food surplus to people in need. Yindii, the first food-sharing platform in Thailand, offers its customers food surplus in "blind boxes". Food delivery services have increased significantly during the pandemic in Thailand, thanks to its high internet penetration (almost 78%) [29] and a significant amount of time spent on smartphones, including food choice [18]. Propelled by the COVID-19 pandemic lockdown restrictions, the number of online food deliveries increased to 12.2 million deliveries in 2022 [29,30]. Yindii and the "blind box" delivery method are the data sources for the present research, warranting a comprehensive introduction before delving into the research objectives.

### 2.1. Yindii

Some food-sharing platforms have been founded in South East Asia, for example, Surplus from Indonesia [31] and Yindii from Thailand. Yindii is a newly founded company that offers food surplus-sharing services and is the first food-sharing platform in Thailand, with more than 100,000 users at present [32]. Yindii provides high-quality unsold food from starred hotels, premium bakeries, and high-end supermarkets with up to 80% discount [33]. To date, Yindii has already cooperated with over 700 partners, rescued more than 110,000 meals, prevented around 275,000 kg of $CO_2$ emissions, and become the No 1 food surplus application in Asia [32]. In this research, Yindii is selected as the food-sharing platform to study consumer purchase intention towards food surplus.

### 2.2. Blind Box

The idea of the blind box was developed in Japan. It refers to boxes with the same exterior packaging but containing different styles of build-in products [34]. The perceived uncertainty of randomness has positively influenced consumers' impulsive purchase intention [35]. The information gap for blind boxes positively raises the emotive antecedents that lead to purchase intention [36]. Due to the rising popularity of blind box products, this sales model has been adopted in various industries, including the food industry. Triggered by recent digital transformation, an increasing number of businesses are interested in the food surplus blind box business model to establish online connections between suppliers and recipients of food surplus aiming to reduce food waste [37]. The food surplus blind box serves as a unique form of novelty consumption, appealing to curiosity and even a sense of "gambling" for young consumers. Undoubtedly, this concept can encourage consumers to purchase the food surplus. In addition, it presents an innovative approach to tackle the food waste issue by effectively repurposing food surplus [38].

Several researchers explored consumer behavioral intention of food surplus on food-sharing platforms [20,22,38]. In a study of surplus food blind box in China, researchers confirmed the associations of perceived playfulness, convenience, and subjective norm with consumer purchase intention while the relationship between perceived risk and purchase intention is unclear [38]. When investigating food surplus-sharing platforms in two European countries (Italy and Germany), Pisoni, Canavesi, and Michelini [20] found that younger people are more likely to use the new digital platform and the economic benefits are more attractive than environmental concerns. In a case study of OLIO, one of the most popular European food surplus-sharing platforms, Mazzucchelli, Gurioli, Graziano, Quacquarelli, and Aouina-Mejri [22] revealed that consumer familiarity, perception of the environment, and social responsibility positively enhance consumer behavioral response to the use of food surplus-sharing platforms.

The current research is the first attempt to study the factors influencing consumers' purchase intention of food surplus through the food-sharing platform in Thailand, an ASEAN country. In order to know how to reduce food waste at the consumer level mediated through online food-sharing platforms, this study aims to explore the impacts of six selected factors (environmental concern, perceived playfulness, social norms, food waste awareness, price consciousness, and food neophobia) on the purchase intention of Thai consumers of food surplus through the food-sharing platform. Moreover, the study

also investigates to what extent these factors can influence consumers' purchase intention of food surplus in Thailand. The research questions of this study are:

RQ1 Do the six stated factors influence Thai consumers' purchase intention of food surplus through the food-sharing platform?

RQ2 To what extent do the six stated factors influence Thai consumers' purchase intention of food surplus through the food-sharing platform?

Several factors have been examined as determinants of purchase intention of food surplus blind boxes in China [38], they are subjective norm, perceived food quality, brand image, perceived playfulness, perceived variety, and convenience. The food from Yindii is high quality from premium bakeries and starred hotels, therefore perceived food quality and brand image are not tested in this study as consumers do not care about the quality and brand when buying Yindii food surplus blind boxes [38]. In a previous study, perceived variety has been found as having no impact on the intention to buy food surplus blind boxes [38]. This factor is not included in this research as well. In addition, as Thai consumers have been familiar with online food delivery services since the pandemic and convenience has been tested as an influential factor impacting the use of food delivery apps both before and during the pandemic in Thailand [39], convenience is also excluded as a food purchase intention factor.

Besides social norm and perceived playfulness, which have been tested in previous research [38], four more factors are tested in the context of Thailand. Environmental concern is one of the drivers of food waste prevention behavior [40], and it has been tested regarding consumer behavioral intention on food-sharing platforms [22]. Food waste awareness is a strong factor when examining sustainable food consumption issues; it positively influences purchase intention of suboptimal products [41,42], oddly shaped food [43], and value-added surplus products [44]. Both environmental concern and food waste awareness are examined in this study. Price consciousness is one of the drivers of food waste prevention behavior [45]. Low price is more important for price-conscious consumers when they purchase products than for non-price-conscious consumers [46]. Previous studies indicated that people tend to use food-sharing platforms due to financial considerations rather than environmental motives [20]. To compare which factor is more influential for Thai food surplus consumers, price consciousness is investigated in this research. Food neophobia is one of the key psychological features when examining consumer acceptance of newly developed products [47]. To test Thai consumers' acceptance of the new type of food surplus blind box, food neophobia is included.

The rest of the paper is structured as follows. Section 3 provides an overview of the factors covered by the existing literature. Then, the conceptual framework and the research hypotheses are explained. Section 4 presents the method that was used in the study, and Section 5 shows the most relevant findings and results from the analysis. Section 6 discusses, compares, and contrasts findings between the current study and previous research. Finally, Section 7 provides the conclusions, research limitations, and suggestions for future research.

## 3. Literature Review

### 3.1. Purchase Intention

Individual intention serves as an indicator of the degree of effort people are willing to exert in order to carry out a behavior [48]. In general, the more intention to engage in a behavior, the more likely it will be carried out [48]. Broekhuizen [49] found that there are no obvious distinctions in the factors' impacts on purchase intention between online and offline contexts. In the context of surplus food, purchase intention is defined as willingness to buy waste-to-value food [47] and oddly shaped food [43]. In the online setting, purchase intention is viewed as a willingness to use a food-sharing platform to buy [22] and willingness to purchase food surplus blind boxes [38]. Following [50,51] for this research, purchase intention is defined as consumers' intention to buy food surplus through food-sharing platforms.

*3.2. Environmental Concern*

Environmental concern encompasses all aspects of an individual's relationship with the environment, including perceptions, emotions, knowledge, attitudes, values, and behaviors. In short, the term describes general attitudes that focus on the goal of environmental protection [52,53]. People are likely to engage in environmental behavior based on their belief about how an object influences their values (environmental concern) [54]. In this paper, environmental concern is defined as individuals' overall attitudes toward the environment when purchasing food surplus from food-sharing platforms [53]. In the context of food consumption, the environmental aspect has been widely discussed in previous studies, and a positive impact of environmental concern on food waste prevention behavior is supported by previous studies [55,56]. Environmental concern has impacts on purchase intention of suboptimal food in China and Egypt [41], food surplus [22], and waste-to-value food in Italy [47]. Using an application to redistribute the food surplus can reduce food waste and improve environmental well-being [21]. Consumer perception of environmental responsibility positively influences consumer behavioral intention to use food-sharing platforms [22]. Similarly, consumers are likely to have a positive purchase intention toward products with more environmental benefits [47]. In this study, consumers may likely to buy food surplus with environmental benefits.

*3.3. Perceived Playfulness*

Perceived playfulness is widely studied as a factor influencing purchase intention in the online and digital shopping context. The playfulness and the benefits of the online shopping experience might be seen as intrinsic motivation [57]. Users with a higher capability to perceive playfulness tend to experience more favorable emotions and a higher level of satisfaction [58]. Perceived playfulness, as an intrinsic motivation factor, strongly influences people's behavioral intentions [59]. Yang, Chen, Sun, Wei, Miao, and Gu [38] define perceived playfulness of food surplus blind boxes as "the degree to which the consumer believes that enjoyment could be derived when shopping for the surplus food blind box". Perceived playfulness plays an important role in influencing impulse purchase intention in the online environment [60].

*3.4. Social Norms*

Social norms substantially influence human behavior [61,62]. They are the main determinant of behavioral intentions in the theory of planned behavior [48]. The use of social norms is a strategy to change behaviors [42]. Equity and social responsibility are two commonly recognized social norms [63]. Social norms comprise two aspects: descriptive norms refer to commonly accepted behaviors, and injunctive social norms refer to behaviors considered morally correct or what ought to be done [61,64]. Vermeir and Verbeke [65] view social norms as "the perceived social pressure to perform or not to perform the behavior". Yang et al. [38] view social norms not just as perceived social pressure but also as the personal motivations to conform to others' views. In the context of food, social norms refer to the influence of peers on consumers' purchase decisions [66] or food waste behavior [67]. Empirical studies support that social norms positively influence consumers' intention [68] to buy suboptimal food [42], sustainable dairy products [65,69], and food surplus blind boxes [38]. Consumers may be confused and hesitate to buy uncertain food, thus, consumers may follow others' behavior.

*3.5. Food Waste Awareness*

In Schwartz's norm activation theory [63] and Stern's value–belief–norm theory [70], environmental problem awareness is an important antecedent to pro-environmental behavior. Schwartz [63] views awareness of consequences as when one is "aware of the consequences of one's behavior for others". Chen [55] views consumers with food waste awareness as people who recognize the negative impacts of food waste. Food waste awareness is a strong factor when examining sustainable food consumption issues, and

it positively influences purchase intention of suboptimal products [41,42], oddly shaped food [43], and value-added surplus products [44]. Some scholars found that food waste awareness indirectly influences suboptimal food purchase intention [71]. However, some did not prove the association [72]. In this study, the awareness of food waste may influence consumers' intention to buy food surplus because it is an effective way to reduce food waste [12].

### 3.6. Price Consciousness

Price has been viewed as a crucial marketing factor that influences consumers' buying behavior [51,73]. In the context of food consumption, the discount rate has a significant impact on consumers' purchase intention toward expiring food [74] and suboptimal food [72]. Price consciousness refers to an individual characteristic that distinguishes consumers according to the degree of importance they put on price when deciding whether or not to buy products [50]. Or it can be defined as "the degree to which the consumer focuses exclusively on paying low prices" [73]. For consumers who are price conscious, low price is more important when they purchase the products than for those who are not price conscious [46]. Price consciousness influences consumers' purchase intention differently toward different products. Price consciousness is one of the determinants of food waste prevention behavior [45]. It positively influences consumers' purchase intention toward expiring dated food [51] and suboptimal foods [19]. On the contrary, some scholars support that price consciousness negatively influences purchase intention for organic food in discounted settings [45] and new food products among high-knowledge consumers [50]. Food surplus drops in the same category of expiring dated food and suboptimal foods. Therefore, in this study, the positive relation between price consciousness and purchase intention is tested.

### 3.7. Food Neophobia

Regarding consumers' acceptance of newly developed food products, individuals generally have an aversion to new foods [47]. It is important to analyze two key psychological features when examining consumer acceptance of newly developed products. Both aspects indicate an aversion to new foods [47]. The first factor is food neophobia, which is defined as when consumers show a strong tendency to avoid trying new foods [75] and unfamiliar food [76]. The second factor is food technology neophobia, which has the potential to influence consumers' attitudes concerning food processed by new methods [77]. In this study, we investigate the food surplus that is not processed in novel ways but just redistributed on a new platform. Therefore, we only pick food neophobia as an influential factor to test. Arvola et al. [78] indicate that food neophobia is not just the tendency to avoid novel food but also novel food aversion. In this study, food neophobia refers to a strong aversion to trying the food surplus through food-sharing platforms. Verbeke [79] found that food neophobia is the most influential factor and a major barrier to consumers' readiness to try novel foods. Food neophobia negatively influences consumers' purchase intention of upcycled food [80] and waste-to-value food [47]. Thus, food neophobia may be a barrier to buying food surplus from Yindii.

### 3.8. Conceptual Framework

The study develops a conceptual framework based on the discussion above. There are several factors, including environmental concern, perceived playfulness, social norms, food waste awareness, price consciousness, and food neophobia, that influence purchase intention (Figure 1). The following hypotheses are developed.

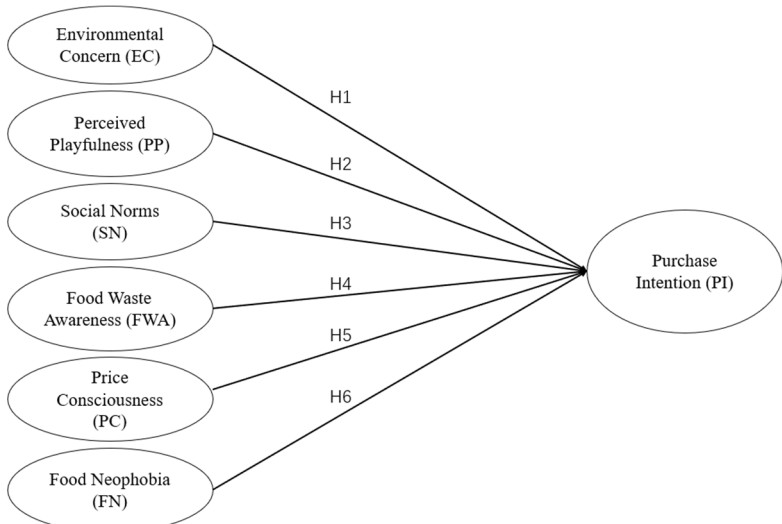

**Figure 1.** Conceptual Framework of the Research.

**Hypothesis 1 (H1).** *Environmental concern positively influences consumers' purchase intention of food surplus through food-sharing platforms.*

**Hypothesis 2 (H2).** *Perceived playfulness positively influences consumers' purchase intention of food surplus through food-sharing platforms.*

**Hypothesis 3 (H3).** *Social norms positively influence consumers' purchase intention of food surplus through food-sharing platforms.*

**Hypothesis 4 (H4).** *Food waste awareness positively influences consumers' purchase intention of food surplus through food-sharing platforms.*

**Hypothesis 5 (H5).** *Price consciousness positively influences consumers' purchase intention of food surplus through food-sharing platforms.*

**Hypothesis 6 (H6).** *Food neophobia negatively influences consumers' purchase intention of food surplus through food-sharing platforms.*

## 4. Materials and Methods

The study employs a quantitative research method using online survey questionnaires in order to gain insights from current food surplus buyers through two research questions: Do the six stated factors influence Thai consumers' purchase intention of food surplus through the food-sharing platform? To what extent do the six stated factors influence Thai consumers' purchase intention of food surplus through the food-sharing platform?

### 4.1. Questionnaire Development

The questionnaire consists of closed-ended questions with three sections. In the first section, three screening questions aim to include targeted samples and filter out invalid respondents. The respondents are residents of Thailand and are users of Yindii. In addition, the sample consists of people who have lived in Thailand for the past six months (2022). The respondents are required to be over 18 years old to be considered mature enough as participants. The participants respond to the questions voluntarily, and they are free to withdraw at any time. All respondents participate in this survey on a voluntary and informed basis.

In the second section, scale questions are the main part of the survey, which reflect each variable including environmental concern, perceived playfulness, social norms, food

waste awareness, price consciousness, food neophobia, and purchase intention. According to the objective of this research, the constructs are adapted from validated scales of previous studies and measured using a five-point Likert scale. All scale questions (Appendix A) are rated on a Likert scale of 1 to 5 (strongly disagree to strongly agree). Environmental concern (EC) questions have five items that are adapted from Kim and Choi [53]. Perceived playfulness (PP) questions have four items derived from Chu and Lu [81]. Scales with five items from Vermeir and Verbeke [65] are employed to measure the subjects' social norms (SN). Scales with four items submitted by Loebnitz, Schuitema, and Grunert [43] are adopted to measure food waste awareness (FWA). For the price consciousness (PC) scale, five measurement items are modified from Konuk [51]. Six measurement items developed by Pliner and Hobden [82] and selected by Verbeke [79] are used to evaluate food neophobia (FN). The scale of purchase intention (PI) consists of three items developed from Konuk [51].

In the third section, demographic information is collected at the end of the survey. Gender, age, educational level, monthly income, and nationality are included in the demographic section.

The questionnaires were developed in English based on previous studies. Then, the English version of the survey was translated into Thai. The back translation technique [83] was applied to ensure the clarity and correctness of the survey in this cross-cultural study. Besides the primary researcher, one English native speaker and four bilingual Thai native speakers in the relevant field reviewed and refined questionnaires. Minor revisions were made in the final version of the survey. The questionnaire was organized in a shuffled manner (the questions were sequenced in a random manner) and five reversed questions were used to reduce the respondents' bias. Prior to the survey distribution, the questionnaire was approved by Mahidol University Central Institutional Review Board.

### 4.2. Data Collection

The online survey was distributed using convenience sampling because this technique is a cost-effective approach to collecting a large number of samples [84]. Research data were collected from 1 April 2023 to 1 May 2023, 30 days in total, by Google Forms. The survey was distributed through the email system of Yindii. In total, 11,631 emails were sent out by Yindii, and 406 respondents participated in the survey. In the end, 284 participants passed all three screening questions. They lived in Thailand in the past six months, are 18 years old or over, and have purchased food surplus through the food-sharing platform at least once in the past six months. With all questions completed, 284 questionnaires (70%) are valid for data analysis. The desired ratio of observations to variables is 20:1 in factor analysis and multiple regression analysis [85]. This research has six independent variables. Therefore, 120 observations are the threshold for data collection. In addition, 200 is a commonly used threshold for major types of market research with a non-probability sampling technique [84]. There is a moderate precision increase beyond a sample size of 200 [86]. Previous empirical studies were analyzed with around 200 samples. Saleki et al. [87] explored six drivers of organic food purchase intention and their influence on purchase behavior with 246 consumers. Zhang, Zhou, and Qin [35] examined four parameters of impulsive purchase intention of blind box products with 193 valid questionnaires. Köpcke [44] tested four hypotheses of value-added surplus products with 201 participants. Although Yang et al. [38] analyzed 15 estimated parameters with 735 valid samples to investigate relevant factors influencing consumers' purchase intention of food surplus blind boxes, they also suggested that the threshold of the ratio of estimated parameters to the number of samples should be 1:10.

### 4.3. Data Analysis

In this research, the Statistical Package for Social Science (SPSS) version 25 is used for quantitative data analysis. Descriptive analysis and inferential analysis, such as *t*-tests and ANOVA, are conducted to describe the demographic information of respondents and

compare the differences among groups. Exploratory factor analysis and multiple regression analysis are employed to examine causal relationships among factors. Varimax rotation is used for factor analysis. Bartlett's test of sphericity and the Kaiser–Meyer–Olkin test are used to test the assumptions prior to carrying out the analysis [85]. FWA 4.3, PC 5.3, FN 6.1, FN 6.4, and FN 6.6 are rotated before factor analysis.

## 5. Results

Initially, 406 respondents participated in the survey, and 284 questionnaires were valid for further data analysis. The data were extracted from Google Forms into an Excel file. Then, a coding process was conducted to prepare data analysis by SPSS. Age group and educational background items were regrouped due to low item percentage (less than 10%). After data screening and cleaning, we analyzed the data by applying descriptive analysis, exploratory factor analysis, regression analysis, a *t*-test, and ANOVA to develop sufficient findings to answer the research questions.

### 5.1. Descriptive Analysis

With a 70% validity rate, among the 284 valid respondents (Table 1), 190 users are female, 82 users are male, and 12 users represent other genders or do not disclose their gender. Consumers aged 18–30 and 31–40 are the main groups who purchase food surplus online, representing 30.6% and 36.6% of the platform users. According to participants' educational backgrounds, the vocational college/diploma and below group includes 17 respondents which represent 6% of the participants and is the smallest group of respondents. Referring to monthly income, respondents are distributed into five groups, among which the THB 60,000 and above group is the biggest group, which accounts for 32.4% of the respondents. By contrast, the lower than THB 15,000 group is the smallest one, which only represents 9.2% of the participants. Regarding nationalities, the main users of Yindii are local Thais (92.6%).

**Table 1.** Demographic profiles of respondents (n = 284).

| Sample | Category | Number | Percentage |
|---|---|---|---|
| Gender | Female | 190 | 66.9% |
| | Male | 82 | 28.9% |
| | Others | 12 | 4.2% |
| Age | 18–30 | 87 | 30.6% |
| | 31–40 | 104 | 36.6% |
| | 41–50 | 67 | 23.6% |
| | 51 and Above | 26 | 9.2% |
| Education Level | Bachelor's Degree | 141 | 49.6% |
| | Master's Degree or Above | 126 | 44.4% |
| | Vocational College/Diploma and Below | 17 | 6.0% |
| Monthly Income | Lower than THB 15,000 | 26 | 9.2% |
| | THB 15,001–30,000 | 62 | 21.8% |
| | THB 30,001–45,000 | 58 | 20.4% |
| | THB 45,001–60,000 | 46 | 16.2% |
| | 60,001 THB and Above | 92 | 32.4% |
| Nationality | No, I am not Thai | 21 | 7.4% |
| | Yes, I am Thai | 263 | 92.6% |

### 5.2. Exploratory Factor Analysis

Bartlett's test of sphericity and the Kaiser–Meyer–Olkin test were used to test the assumptions prior to carrying out factor analysis. In Bartlett's test of sphericity, the *p*-value is less than 0.05. Therefore, there are correlations among factors. Furthermore, the Kaiser–Meyer–Olkin test demonstrates that KMO (0.871) is greater than 0.5 and the model is suitable for analysis [85]. According to Table 2, six factors' eigenvalues are greater than

1 [85]. In total, those six factors can explain 60.6% of the variance in the original data. A commonly used guideline for this criterion is that 60 percent of overall variance should be considered [88]. In the factor solution, 60.6% exceeds this criterion. The six factors can explain 26.0%, 10.0%, 8.2%, 6.7%, 5.1%, and 4.4% of the total variance, respectively.

**Table 2.** Eigenvalues.

| Component | Eigenvalue | % of Variance | Cumulative % |
| --- | --- | --- | --- |
| 1 | 7.297 | 26.062 | 26.1 |
| 2 | 2.806 | 10.023 | 36.1 |
| 3 | 2.303 | 8.225 | 44.3 |
| 4 | 1.866 | 6.666 | 51.0 |
| 5 | 1.441 | 5.146 | 56.1 |
| 6 | 1.253 | 4.473 | 60.6 |
| 7 | 0.964 | 3.442 | 64.0 |

Table 3 demonstrates the factor loadings and the uniqueness of all items. It is clear that all factor loadings are larger than 0.5, no cross-loading existed, and all uniquenesses are lower than 0.6 [85]. Therefore, no items were removed from the original questionnaire. Six groups of items were presented.

**Table 3.** Component Matrix.

| | 1 | 2 | 3 | 4 | 5 | 6 | Uniqueness |
| --- | --- | --- | --- | --- | --- | --- | --- |
| SN 3.1 | 0.810 | | | | | | 0.258 |
| SN 3.5 | 0.789 | | | | | | 0.312 |
| SN 3.4 | 0.758 | | | | | | 0.347 |
| SN 3.2 | 0.755 | | | | | | 0.322 |
| SN 3.3 | 0.701 | | | | | | 0.418 |
| EC 1.2 | | 0.751 | | | | | 0.396 |
| EC 1.3 | | 0.732 | | | | | 0.392 |
| EC 1.4 | | 0.699 | | | | | 0.449 |
| EC 1.5 | | 0.694 | | | | | 0.394 |
| EC 1.1 | | 0.691 | | | | | 0.473 |
| FWA 4.1 | | 0.611 | | | | | 0.512 |
| PP 2.4 | | | 0.760 | | | | 0.269 |
| PP 2.2 | | | 0.700 | | | | 0.376 |
| PP 2.1 | | | 0.679 | | | | 0.386 |
| PP 2.3 | | | 0.675 | | | | 0.461 |
| FWA 4.2 | | | 0.642 | | | | 0.439 |
| PC 5.4 | | | | 0.841 | | | 0.220 |
| PC 5.2 | | | | 0.816 | | | 0.291 |
| PC 5.1 | | | | 0.758 | | | 0.331 |
| PC 5.5 | | | | 0.509 | | | 0.584 |
| FN 6.5 | | | | | 0.759 | | 0.291 |
| FN 6.2 | | | | | 0.752 | | 0.398 |
| FN 6.3 | | | | | 0.739 | | 0.406 |
| PC 5.3 | | | | | −0.583 | | 0.523 |
| FN 6.6 | | | | | | 0.692 | 0.400 |
| FN 6.1 | | | | | | 0.654 | 0.436 |
| FN 6.4 | | | | | | 0.626 | 0.425 |
| FWA 4.3 | | | | | | 0.555 | 0.527 |

The first group contains six items (SN 3.1, SN 3.2, SN 3.3, SN 3.4, SN 3.5) that illustrate social norms that influence consumers' purchase intention. The second group involves six items (EC 1.1, EC 1.2, EC 1.3, EC 1.4, EC 1.5, FWA 4.1) that are associated with environmental concern, which people consider when they purchase food. The third group comprises five items (PP 2.1, PP 2.2, PP 2.3, PP 2.4, FWA 4.2) that explain the perceived playfulness,

meaning the food-sharing platform users have fun and feel excited when buying through food-sharing platforms. The fourth group includes four items (PC 5.1, PC 5.2, PC 5.4, PC 5.5) that represent price consciousness of the food surplus buyers. The construct explains whether consumers consider the importance of the price when purchasing food surplus. Four items are included in the fifth group (FN 6.2, FN 6.3, FN 6.5, PC 5.3). The items explain food neophobia, a reluctance to try novel flavors or new food on food-sharing platforms. Four items are included in the sixth group (FN 6.1, FN 6.4, FN 6.6, FWA 4.3), illustrating customers' positive attitudes when purchasing novel food through food-sharing platforms. Overall, based on the latent root criterion, the percentage of the variance criterion, and the factor loadings criterion [85], six factors were revealed from exploratory factor analysis. Those factors are social norms, environmental concern, perceived playfulness, price consciousness, food neophobia, and food neophilia.

*5.3. Regression Analysis*

In the regression test, we examine the linear relationships between six independent variables, namely environmental concern, perceived playfulness, social norms, price consciousness, food neophobia, and food neophilia, and one dependent variable, purchase intention.

We can see the mean score for each factor in Table 4. A mean score of 1 indicates "totally disagree". By contrast, a mean score of 5 indicates "totally agree". Participants show strong positive opinions on environmental concern and perceived playfulness with a mean score of 4.37 and 4.18. The mean scores of social norms (3.45), price consciousness (3.9), and food neophilia (3.76) illustrate a medium level of agreement. There is only one mean score less than 3, at 2.76 (food neophobia). It means that the respondents have a negative view of this factor. Environmental concern (0.829), social norms (0.876), and perceived playfulness (0.832) have strong reliability with a high Cronbach's $\alpha$. Price consciousness (0.787) and food neophobia (0.708) have moderate reliability with a medium Cronbach's $\alpha$. While food neophilia has weak reliability with the lowest Cronbach's $\alpha$ (0.613), it is still larger than 0.600 [85], which is acceptable.

**Table 4.** Mean Score and Reliability.

|  | Environmental Concern | Perceived Playfulness | Social Norms | Price Consciousness | Food Neophobia | Food Neophilia |
|---|---|---|---|---|---|---|
| N | 284 | 284 | 284 | 284 | 284 | 284 |
| Mean | 4.37 | 4.18 | 3.45 | 3.90 | 2.76 | 3.76 |
| SD | 0.588 | 0.640 | 0.890 | 0.773 | 0.866 | 0.768 |
| $\alpha$ | 0.829 | 0.832 | 0.876 | 0.787 | 0.708 | 0.613 |

Note: SD = Standard Deviation; $\alpha$ = Cronbach's $\alpha$.

Regarding the model fit test, $p < 0.05$, and this model is suitable to explain the dependent variable. Adjusted $R^2 = 0.478$ indicates that this model can explain 47.8% of the change in the dependent variable (purchase intention). According to Table 5, environmental concern ($p < 0.001$) and perceived playfulness ($p < 0.001$) significantly influence purchase intention. H1 and H2 are supported. Other factors, such as social norms ($p = 0.188$), price consciousness ($p = 0.768$), food neophobia ($p = 0.138$), and food neophilia ($p = 0.710$), do not significantly influence purchase intention. H3, H5, and H6 are not supported. According to the output of exploratory factor analysis, a new factor, food neophilia, emerged. Thus, in this study, we show a new connection between food neophilia and purchase intention of food surplus.

In this model, the results of multiple regression (Figure 2) illustrate the relationships between environmental concern, perceived playfulness, social norms, price consciousness, food neophobia, food neophilia, and purchase intention. Environmental concern (standardized coefficient $\beta = 0.254$) and perceived playfulness (standardized coefficient $\beta = 0.565$) positively influence purchase intention. The high mean scores also show consumers' prefer-

ences toward environmental concern (mean = 4.37) and perceived playfulness (mean = 4.18). For each increase in perceived playfulness and environmental concern by one unit each, purchase intention increases by 0.565 units and 0.254 units. Other factors do not show a relation to purchase intention.

**Table 5.** Model Coefficients.

| Relationships | SE | *t* | *p* | Stand. Estimate | Results |
|---|---|---|---|---|---|
| Environmental Concern—Purchase Intention | 0.0628 | 4.985 | <0.001 | 0.2538 | Supported |
| Perceived Playfulness—Purchase Intention | 0.0647 | 9.900 | <0.001 | 0.5645 | Supported |
| Social Norms—Purchase Intention | 0.0445 | −1.320 | 0.188 | −0.0720 | Not Supported |
| Price Consciousness—Purchase Intention | 0.0459 | −0.295 | 0.768 | −0.0145 | Not Supported |
| Food Neophobia—Purchase Intention | 0.0373 | −1.488 | 0.138 | −0.0663 | Not Supported |
| Food Neophilia—Purchase Intention | 0.0456 | 0.373 | 0.710 | 0.0180 | Not Supported |

Note: Stand. Estimate = Standard Estimate.

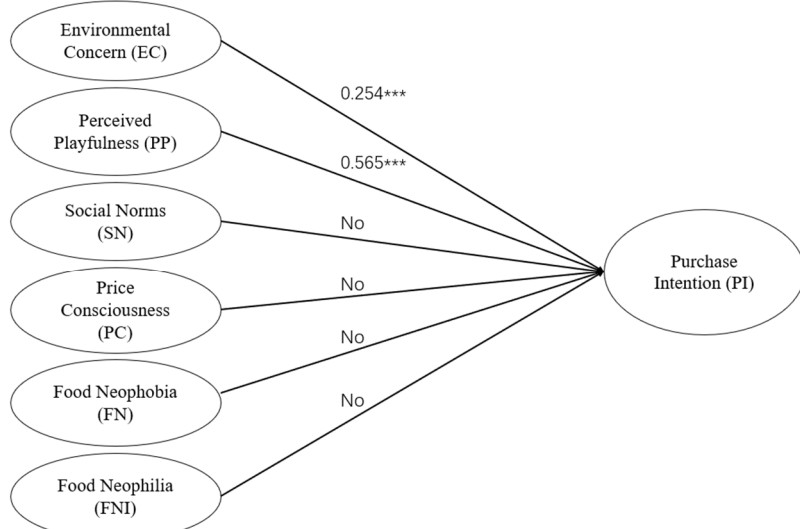

**Figure 2.** Results of Multiple Regression. Note: *** stands for significant at <0.001.

*5.4. t-Test*

A *t*-test can be used to test whether the means from two independent groups are the same or not [88]. In this research, we test whether the means of purchase intention of food surplus through food-sharing platforms are the same between Thai and non-Thai respondents. According to the results, the figures ($p$ = 0.104 > 0.05) indicate that there is no significant difference between the means of purchase intention of the two groups. Therefore, there is no significant difference in purchase intention of food surplus through food-sharing platforms between Thai and non-Thai groups. All consumers share similar attitudes toward food surplus purchase intention.

*5.5. ANOVA*

ANOVA is applied to test the statistical difference between the means of two or more groups [88]. In this study, we used one-way ANOVA to test whether there is a significant difference among various categorical variables, such as gender, age group, educational background, and monthly income. According to the ANOVA, there is no statistical difference in purchase intention among different gender groups ($p$ = 0.066 > 0.05) and age groups ($p$ = 0.849 > 0.05). Consumers from different gender and age groups show the same attitudes to food surplus purchase intention.

There is a statistical difference in purchase intention among different education groups ($p$ = 0.037 < 0.05). To discover which two groups are significantly different, we applied a post hoc test with Bonferroni correction (Table 6).

**Table 6.** Post Hoc Test—Education Level.

| Education Level | Education Level | Mean Difference | SE | df | t | p$_{bonferroni}$ | Cohen's d |
|---|---|---|---|---|---|---|---|
| Bachelor's Degree | Master's Degree or Above | −0.050 | 0.088 | 281 | −0.570 | 1.000 | −0.070 |
| | Vocational College/Diploma and Below | 0.431 | 0.184 | 281 | 2.331 | 0.061 | 0.598 |
| Master's Degree or Above | Vocational College/Diploma and Below | 0.481 | 0.185 | 281 | 2.587 | 0.031 | 0.668 |

We can state that there is a significant difference in purchase intention of food surplus through food-sharing platforms between participants with the educational background of master's degree or above and vocational college/diploma and below ($p = 0.031 < 0.05$). In addition, the mean difference is 0.481, which shows that the participants with a master's degree or above have a higher purchase intention than the respondents with vocational college/diploma and below. Consumers with higher education levels have a stronger purchase intention for food surplus.

According to the ANOVA, there is a significant difference in purchase intention between different participants with different monthly incomes ($p = 0.013 < 0.05$). To discover which two groups are significantly different, we applied a post hoc test with Bonferroni correction.

Referring to Table 7, there is a significant difference in purchase intention between monthly income of THB 15,001–30,000 and lower than THB 15,000 ($p = 0.035$) and between THB 60,000 and above and lower than THB 15,000 ($p = 0.029$). Consumers with lower incomes have relatively higher purchase intention for food surplus. The mean difference between THB 15,000–30,000 and lower than THB 15,000 is −0.4913. The mean difference between THB 60,001 and above and lower than THB 15,000 is −0.4766. Furthermore, the mean differences between all income groups and the lower than THB 15,000 group are negative. It means that the group with monthly income lower than THB 15,000 has a relatively higher purchase intention of food surplus through food-sharing platforms.

**Table 7.** Post Hoc Test—Monthly Income.

| Monthly Income | Monthly Income | Mean Difference | SE | df | t | p$_{bonferroni}$ | Cohen's d |
|---|---|---|---|---|---|---|---|
| THB 15,001–30,000 | THB 30,001–45,000 | −0.1969 | 0.130 | 279 | −1.509 | 1.000 | −0.2757 |
| | THB 45,001–60,000 | −0.2539 | 0.139 | 279 | −1.827 | 0.688 | −0.3554 |
| | THB 60,001 and Above | −0.0147 | 0.117 | 279 | −0.125 | 1.000 | −0.0206 |
| | Lower than THB 15,000 | −0.4913 | 0.167 | 279 | −2.944 | 0.035 | −0.6879 |
| THB 30,001–45,000 | THB 45,001–60,000 | −0.0570 | 0.141 | 279 | −0.404 | 1.000 | −0.0798 |
| | THB 60,001 and Above | 0.1822 | 0.120 | 279 | 1.521 | 1.000 | 0.2551 |
| | Lower than THB 15,000 | −0.2944 | 0.169 | 279 | −1.747 | 0.818 | −0.4122 |
| THB 45,001–60,000 | THB 60,001 and Above | 0.2391 | 0.129 | 279 | 1.854 | 0.648 | 0.3348 |
| | Lower than THB 15,000 | −0.2375 | 0.175 | 279 | −1.355 | 1.000 | −0.3325 |
| THB 60,001 and Above | Lower than THB 15,000 | −0.4766 | 0.159 | 279 | −3.004 | 0.029 | −0.6673 |

## 6. Discussion

In this section, we interpret the findings from data analysis and compare the findings between the current research and the results from previous articles. According to the original conceptual framework, this study tests the relationships between six factors and one dependent variable, purchase intention. The final model only tests five hypotheses, and a new factor, food neophilia, is added to the framework.

### 6.1. Environmental Concern

The attitudes toward the environment when purchasing food surplus through food-sharing platforms positively influence consumers' purchase intention. It means that the more consumers are concerned about the environment the more they are likely to purchase

food surplus through food-sharing platforms. The finding is aligned with the previous studies which mentioned that environmental concern positively influences purchase intention for green products [53,89], use of food-sharing platforms [22], and engagement in food waste-reducing behavior [55]. Similarly, products with more environmental benefits positively lead to higher purchase intention for waste-to-value food [47]. Environmental concern could be an effective aspect to promote food-sharing platforms by policymakers and retailers. Managers could highlight the environmental benefits of the food-sharing platform as selling points to appeal to environmentally friendly consumers.

### 6.2. Perceived Playfulness

The degree of enjoyment in purchasing food surplus through food-sharing platforms influences consumers' purchase intention. The more consumers enjoy the shopping experience, the more they are likely to purchase food surplus on the platform. The association between the factors is supported by other researchers. Chen, Yeh, and Lo [60], Fu and Liang [58], and Kim and Jun [90] also state that perceived playfulness positively influences purchase intention when consumers shop online. Consumers are more interested in their overall experience than the food in a blind box [38]. Making blind boxes more interesting may be an effective way to maintain current users. Fun and enjoyment could be the main selling points for future food-sharing businesses.

### 6.3. Other Factors

Social norms, food waste awareness, price consciousness, food neophobia, and food neophilia do not have impacts on purchase intention in this study. The negligible role of social norms on purchase intention is supported in buying organic food in India [91] and in China [92] and organic clothing in India [93]. The weak role of social norms may be because the early buyers do not have role models to refer to even in collectivist emerging countries [91–93]. In this study, the correlation between food waste awareness and purchase intention could not be tested, because food waste awareness did not form a construct from exploratory factor analysis. Therefore, H4 is rejected. de Hooge, Oostindjer, Aschemann-Witzel, Normann, Loose, and Almli [72] also do not prove the associations between the factors. The limitation on questionnaire development and translation may be the major cause of this issue. In this study, consumers' intention to buy food surplus through food-sharing platforms is impacted by environmental concerns instead of price issues. Tsalis [94] also found that price consciousness does not have an effect on purchase intention of suboptimal food in many European countries. Communicating budget savings [95] and solely the cheap price is insufficient to encourage consumers to buy suboptimal food [94].

Findings from this study align with the results that food neophobia is not an effective factor to influence purchase intention for novel food [78,96]. Food neophilia is viewed as the opposite concept of food neophobia [97]. Thai consumers' consumption of edible insects is highly influenced by neophilic tendency [98]. However, there is no causal relationship between food neophilia and purchase intention of food surplus. Both food neophobia and food neophilia are rejected probably because Thai consumers are already familiar with online food delivery due to the pandemic [39] and the food surplus from starred hotels and premium bakeries is not extremely risky and novel [78].

### 6.4. Demographic Characteristics

The main consumers of the Yindii application are local Thais. There are noticeably more female users than male users, which is the opposite of the demographic distribution in China [38]. The significant gender inequality could be explained by qualitative studies in the future. Young generations (under 40) [38,99] are food surplus-preferring consumers, probably because of their quick adoption of novel food and they show a greater preference for sustainable products [99]. People who have received higher education (bachelor's degree or above) are major food surplus buyers. While Yindii users are distributed widely across various income segmentations, the low-income group shows a higher purchase

intention toward food surplus products. The high-income group, those who earn more than THB 60,000, accounts for over 30% of the total consumers and is the major target group. The same income group only represents around 5% of the total users of food delivery apps in Thailand [39]. The noticeable gap between studies is expected to be explored.

## 7. Conclusions, Limitations, Future Research, and Recommendations

Food is the most vital daily product in our life. A large amount of the population still cannot access enough food. However, a substantial amount of food is lost or wasted every year. To address these issues, food-sharing platforms have been developed to redistribute food surplus. This paper investigates the factors influencing purchase intention of food surplus through a food-sharing platform in Thailand. Results indicate that perceived playfulness is the most influential factor influencing Thai consumers' purchase intention of food surplus, followed by environmental concern. Although the current research was sophisticatedly developed, there are still several limitations. First of all, the cultural differences and the difficulty of the Thai language may have impacted respondents' understanding of the original questions, which may be why the food waste awareness construct was not shown in the result of exploratory factor analysis. Secondly, the researchers collected data from a single food-sharing platform, Yindii, which is the biggest and most successful food-sharing platform in Thailand. The majority of the consumers are from the Bangkok region due to the service coverage, and the generalizability may be impacted by the single data source. Thirdly, Cronbach's alpha of food neophilia is lower than 0.7 (0.613) but greater than 0.6. This may be because the food neophobia scale was a bipolar scale to assess both food neophobia and neophilia at the same time [100]. A clear cut-off point or standardized approach could be tested in a future study to avoid this issue.

### 7.1. Future Research

This study is the first attempt to examine the factors influencing the purchase intention of food surplus through food-sharing platforms in Thailand. The paper shed light on this topic in the Thai context and South East Asian countries. Several future research directions are proposed in this paper. First of all, the research only selected factors influencing food surplus purchase intention. Future research should examine different potential factors. In addition, qualitative studies could be used to explore potential determinants of consumers' intention to buy food surplus through food-sharing platforms. Furthermore, due to the single data source of Yindii being used for this research, other emerging platforms like Oho could be used for data collection. To generalize the findings, replicating an empirical study with multiple cases in different Asian countries is suggested. Lastly, due to the wide cultural gap between Asian and Western nations, a comparative study can be conducted to compare and contrast the similarities and differences between consumers in different countries.

### 7.2. Recommendations

Based on the findings of this paper, the following recommendations are proposed to policymakers and practitioners in the field of food surplus redistribution not only in Thailand but also in other countries. Policymakers should see food-sharing platforms as an effective tool to address the food waste issue and identify and help with malnutrition. Government authorities are suggested to promote the food surplus and food-sharing concept with various stakeholders, such as supermarkets, hotels, bakeries, and restaurants, in order to attract more relevant businesses to the program. They are also encouraged to develop and implement new rules and regulations in favor of food-sharing platforms and educate the general public and potential consumers about these newly introduced products. The findings show that environmental concern and perceived playfulness considerably impact consumers' purchase intention. Managers should explore and promote how food-sharing systems reduce food waste and benefit the environment to attract consumers who are concerned about our planet. The blind box strategy does play a key role in food-sharing platforms, with its most influential feature: playfulness. Managers should discover how to

develop a more joyful and interesting buying experience within platforms and a blind box strategy. In addition, the price element is not a key determinant that is considered when buying food surplus. Businesses may not need to spend the majority of their time and resources on how to offer discounts. Companies may make efforts to attract and maintain female locals and high-income groups that are the major food surplus consumers.

**Author Contributions:** Conceptualization, N.H., R.S., M.H. and G.P.M.; methodology, N.H.; software, N.H.; validation, N.H., R.S., M.H. and G.P.M.; formal analysis, N.H.; investigation, N.H.; resources, N.H.; data curation, N.H.; writing—original draft preparation, N.H.; writing—review and editing, N.H., R.S., M.H. and G.P.M.; visualization, N.H.; supervision, R.S.; project administration, N.H. All authors have read and agreed to the published version of the manuscript.

**Funding:** This research received no funding.

**Institutional Review Board Statement:** The study was conducted in accordance with the Declaration of Helsinki and approved by the Institutional Review Board (or Ethics Committee) of Mahidol University (protocol code MU-CIRB 2023/105.2403, 24 April 2023).

**Informed Consent Statement:** Informed consent was obtained from all subjects involved in the study.

**Data Availability Statement:** Data are available upon reasonable request.

**Conflicts of Interest:** The authors declare no conflict of interest. The funders had no role in the design of the study; in the collection, analyses, or interpretation of data; in the writing of manuscript, or in the decision to publish the results.

## Appendix A

**Table A1.** Measurement Items.

| Construct | | Item | References |
|---|---|---|---|
| Environmental Concern | EC 1.1 | I am extremely worried about the state of the world's environment and what it will mean for my future. | Kim [53] |
| | EC 1.2 | Mankind is severely abusing the environment. | |
| | EC 1.3 | When humans interfere with nature it often produces disastrous consequences. | |
| | EC 1.4 | The balance of nature is very delicate and easily upset. | |
| | EC 1.5 | Humans must live in harmony with nature in order to survive. | |
| Perceived Playfulness | PP 2.1 | I enjoy the course of purchasing food surplus through Yindii. | Chu [81] |
| | PP 2.2 | Purchase food surplus through Yindii makes me feel pleasant. | |
| | PP 2.3 | When purchasing food surplus through Yindii, I feel excited. | |
| | PP 2.4 | Overall, I found purchasing food surplus through Yindii is interesting. | |
| Social Norms | SN 3.1 | People who are important to me think I should buy food surplus from Yindii. | Vermeir [65] |
| | SN 3.2 | My family thinks I should buy food surplus from Yindii. | |
| | SN 3.3 | Society thinks I should buy food surplus from Yindii. | |
| | SN 3.4 | My friends think I should buy food surplus from Yindii. | |
| | SN 3.5 | People who influence my buying behavior think I should buy food surplus from Yindii. | |
| Food Waste Awareness | FWA 4.1 | Food waste increases the burden on the environment. | Loebnitz [43] |
| | FWA 4.2 | We can avoid food waste by buying food surplus through Yindii. | |
| | FWA 4.3 | It is a good thing that food surplus is not being sold in regular shops. | |
| | FWA 4.4 | Most food surpluses are wasted. | |

**Table A1.** *Cont.*

| Construct | | Item | References |
|---|---|---|---|
| Price Consciousness | PC 5.1 | I am willing to go to extra effort to find lower prices. | Konuk [51] |
| | PC 5.2 | I will grocery shop at more than one store to take advantage of low prices. | |
| | PC 5.3 | The money saved by finding low prices is usually not worth the time and effort. | |
| | PC 5.4 | I would shop at more than one store to find low prices. | |
| | PC 5.5 | The time it takes to find low prices is usually worth the effort. | |
| Food Neophobia | FN 6.1 | I am constantly sampling new and different foods. | Verbeke [79] |
| | FN 6.2 | I don't trust new foods. | |
| | FN 6.3 | If I don't know what is in a food, I won't try it. | |
| | FN 6.4 | At dinner parties, I will try a new food. | |
| | FN 6.5 | I am afraid to eat things I have never had before. | |
| | FN 6.6 | I will eat almost anything. | |
| Purchase Intention | PI 7.1 | I am willing to buy food surplus through Yindii in the future. | Konuk [51] |
| | PI 7.2 | I plan to purchase food surplus through Yindii. | |
| | PI 7.3 | I will make effort to buy food surplus through Yindii. | |

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
