# Peer review of "Factors Influencing Purchase Intention of Food Surplus through a Food-Sharing Platform"

_sustainability, doi:10.3390/su151713000_

Round 1
Reviewer 1 Report
My overall evaluation of the manuscript is positive. There are a number of minor revisions that should be addressed.
1. It seems necessary that a part of the article about diseases that have the ability to become a pandemic, such as COVID-19, was discussed as an environmental concern and its effect in Food Surplus through the Food Sharing Platform.
2. It would have been better to include information about some of the participants' life habits that affect the use of platforms in Table No. 1.
Author Response
Based on the reviewer's recommendations, we made a complete overhaul of the paper. Our apologies, but we hope this helps facilitate a better paper overall.
Sincerely,
The Authors

Reviewer 2 Report
The article is interesting and up-to-date, but some aspects can be improved.
The title of the article goes beyond the content of the study. You analyzed one case study…
Introductions you finish with implications, and recommendations. It should be at the end of the article.
How did you identify the factors, which make an impact on consumers’ purchase intention of food surplus through food-sharing platforms? Why did you choose these particular factors, rejecting others? Please justify your choice.
Consumers who want to buy on food-sharing platforms first should have a positive approach to the technologies. Consumers can use platforms online when their digital literacy is appropriate and when they have a positive attitude toward platforms. How did you evaluate this aspect?
Such statements as “some researchers”, and “some scholars” don’t provide value to this article.
In your model (Figure 1) you defined a dependent variable as “Purchase intention”, but according to the article you want to evaluate “purchase intention of food surplus through the food-sharing platform“. Are there any differences between consumers' intentions online and offline?
Please justify why you think that sample size is appropriate. Please provide a formula, how did you count the appropriate sample?
To which group of users did you make the conclusions?
The hypothesis should be better justified.
Demographic characteristics can be presented in a shorter way.
In your empirical study, you identified six factors, but the hypothesis was tested with seven factors.
„six factors were revealed from factor analysis. Those factors are social norms, environmental concern, perceived playfulness, price consciousness, food neophobia, and food neophilia“?
You have to rethink the results of this factor „Food Waste Awareness“.
Your ambition was to evaluate „which factors have a stronger influence on consumers’ purchase intention“. Please compare these factors and make conclusions/suggestions according to the results.
Can be improved.
Author Response

(The authors gave the same response as above.)

Reviewer 3 Report
The article entitled "Factors influencing the intention to purchase surplus food through the Food Sharing platform" presents interesting information on how various factors influence the intention to purchase surplus food on a food sharing platform. I believe that the article should be restructured in some sections and some comments should be addressed. Below are my observations:
There are several ideas or paragraphs repeated throughout the manuscript.
Lines 121-135: This information should be placed in the discussion or theoretical and practical implications of the research.
Section 2.3 should briefly describe the relationship of the blind box to the surplus feed.
A connecting statement is missing for each of the sections of the literature review, with the proposed hypotheses.
The authors should provide a more complete description of the statistical analyses performed.
Lines 418 - 447. This section should be summarized and the results better discussed.
Lines 463-468. This information should be described in the methodology of the article. What method of variance extraction was used? were the factors rotated?
Were the items of the food neophobia scale construct reversed according to the methodological processes? it should be clarified in the methodology.
Lines 540 - 545: repeated information, suggested to remove
Lines 702 - 726: this information has already been presented in the results, the authors should discuss their findings in the light of different studies.
The conclusions should be rephrased, the authors continue to describe the results of their research.
English proofreading should be carried out. For example, the methodology should be written in the past tense. There are several typing errors.
Author Response

(The authors gave the same response as above.)

Round 2
Reviewer 2 Report
Thank you for the improvement.
The title of the article is too broad.
Minor editing.
